# Effect of Forest Therapy on Depression and Anxiety: A Systematic Review and Meta-Analysis

**DOI:** 10.3390/ijerph182312685

**Published:** 2021-12-01

**Authors:** Poung-Sik Yeon, Jin-Young Jeon, Myeong-Seo Jung, Gyeong-Min Min, Ga-Yeon Kim, Kyung-Mi Han, Min-Ja Shin, Seong-Hee Jo, Jin-Gun Kim, Won-Sop Shin

**Affiliations:** 1Department of Forest Sciences, Chungbuk National University, Cheongju 28644, Korea; well@chungbuk.ac.kr; 2Graduated Department of Forest Therapy, Chungbuk National University, Cheongju 28644, Korea; forest-bb@naver.com (J.-Y.J.); myeongseo@chungbuk.ac.kr (M.-S.J.); akwjs5019@chungbuk.ac.kr (G.-M.M.); rkdus6520@naver.com (G.-Y.K.); gmhan21@gmail.com (K.-M.H.); yeamolove@hanmail.net (M.-J.S.); 3National Center for Forest Therapy, Yeongju 36043, Korea; jorent@chungbuk.ac.kr; 4Korea Forest Therapy Forum Incorporated Association, Cheongju 28644, Korea

**Keywords:** forest therapy, depression, anxiety, meta-analysis

## Abstract

This systematic review and meta-analysis aimed to summarize the effects of forest therapy on depression and anxiety using data obtained from randomized controlled trials (RCTs) and quasi-experimental studies. We searched SCOPUS, PubMed, MEDLINE(EBSCO), Web of science, Embase, Korean Studies Information Service System, Research Information Sharing Service, and DBpia to identify relevant studies published from January 1990 to December 2020 and identified 20 relevant studies for the synthesis. The methodological quality of eligible primary studies was assessed by ROB 2.0 and ROBINS-I. Most primary studies were conducted in the Republic of Korea except for one study in Poland. Overall, forest therapy significantly improved depression (Hedges’s g = 1.133; 95% confidence interval (CI): −1.491 to −0.775) and anxiety (Hedges’s g = 1.715; 95% CI: −2.519 to −0.912). The quality assessment resulted in five RCTs that raised potential concerns in three and high risk in two. Fifteen quasi-experimental studies raised high for nine quasi-experimental studies and moderate for six studies. In conclusion, forest therapy is preventive management and non-pharmacologic treatment to improve depression and anxiety. However, the included studies lacked methodological rigor and required more comprehensive geographic application. Future research needs to determine optimal forest characteristics and systematic activities that can maximize the improvement of depression and anxiety.

## 1. Introduction

Depression and anxiety are considered prevalent mental health problems worldwide. According to the World Health Organization [1], it is estimated that 4.4% of the world’s population suffers from depression, and 3.6% suffer from anxiety disorders. Namely, depression affects about 300 million people, and anxiety affects about 264 million people worldwide.

Recently, widespread psychological consequences of the coronavirus pandemic have been observed at individual, community, national, and international levels [2]. At a personal level, people are more likely to feel sick, dead, or helpless from the coronavirus infection and experience fear from lockdown-induced quarantine. As a result, mental health problems such as depression and anxiety are becoming more serious. For example, Salari et al. [3] performed a meta-analysis on the effects of COVID-19 on the spread of stress, anxiety, and depression. As a result, in five studies with a total sample size of 9074, the stress prevalence was 29.6% (95% CI: 24.3–35.4), in 17 studies with a sample size of 63,439, the anxiety prevalence was 31.9% (95% CI: 27.5–36.7), and in 14 studies with a sample size of 44,531, depression was 33.7% (95% CI: 27.5–4%.6%). In particular, common symptoms of depression can adversely affect health conditions due to sad moods, anxiety, insomnia, loss of vitality, and lack of interest in life [4]. In the worst case can lead to suicide.

Moreover, depression and anxiety are very closely related, and the coexistence rate diagnosed simultaneously is high. It has been reported that approximately 85% of patients with depression experience significant anxiety symptoms, while comorbid depression occurs in up to 90% of patients with anxiety disorder [5]. The coexistence of the two diseases increases the risk of suicide, so treatment of depression and anxiety and preventive management is crucial for public health.

Common treatments for depression and anxiety are pharmacotherapies, such as antidepressants and anti-anxiety drugs. They have advantages such as treatment accessibility and have been proven to improve depression and anxiety symptoms, but there are several disadvantages [6]. For example, the use of antidepressants can have secondary effects such as hypotension and constipation [7], decreased sexual function [8], gastrointestinal symptoms, weight gain, and metabolic abnormalities [9]. Additionally, the anti-anxiety drug can come with side effects such as insomnia, diarrhea, headaches, nausea, jitteriness, or restlessness [10,11]. In addition, pharmacological treatment has potential adverse effects such as the risk of dependence [12,13] and withdrawal symptoms [14,15]. Considering these disadvantages, non-pharmacological treatment can be performed. There is an abundance of evidence on the effectiveness of non-pharmacological intervention for depression and anxiety. The intervention included components of mindfulness-based therapy [16,17], cognitive behavioral therapy (CBT) [18,19], exercise [20,21], and yoga [22,23].

One of these non-pharmacological interventions is direct contact with nature. Human health benefits from exposure to forests vary and include recovery ability such as stress reduction and mental health improvement [24,25]. Forest therapy, also known as “forest bathing,” is a collection of activities to improve human health or welfare in a forest environment. There is quite a variety of methods applied to forest therapy. The critical element of forest therapy is recognition in the forest environment, including the five senses, which can be combined with meditation, forest walking, various recreational activities, and cognitive behavioral therapy [26].

In recent years, forest therapy and its estimated preventive effect are attracting more and more attention. Many previous studies have reported the positive effects of forest therapy on physiological and psychological health. For example, in terms of the physiological effects of forest therapy, previous studies have shown that forest therapy improves immune function by enhancing the activity of NK cells [27], lowering the concentration of cortisol which is a stress hormone [28,29], and balance the autonomic nervous system [30,31,32].

In terms of the psychological effect of forest therapy, it has been reported that forest therapy reduces psychological stress or mental fatigue and induces positive emotions. For example, Morita et al. [33] reported that staying and walking in the forest reduces hostility and depression, and further studies have shown that participants’ anxiety decreased. Furthermore, the amount of sleep improved after walking in the forest [34]. A study by Dolling et al. [35] of middle-aged people reported that both the group that conducted activities in the forest environment and the group that conducted indoor handicraft activities reduced fatigue and stress, and the self-health check-up score increased. Bielinis et al. [36] investigated the psychological effects of the forest environment by dividing the forest environment exposure group and the urban environment exposure group in winter for 62 college students. The results of the study showed that the interaction with the forest in winter had a significantly positive effect on the participants’ emotional and psychological recovery and vitality.

In addition, many studies have shown that natural environments such as forests positively affect mood states [37,38,39]. For example, Pretty et al. [37] reported that the participants’ mood and self-esteem improved considerably after the forest exercise. Joung et al. [38] investigated physiological and psychological reactions using near-infrared spectroscopy. The study results showed a more stable brain condition when viewing the forest landscape than the urban landscape, and negative sub-factors such as anger and fatigue were low, while vitality was high. A study by Song et al. [39] investigated female college students’ physiological and psychological effects while looking at the forest landscape while comparing exposure to the urban context and its impact. The results reported that looking at the forest landscape significantly reduces participants’ negative emotions and anxiety and increases positive emotions compared to exposure to the urban environment. Triguero-Mas et al. [40] also reported that when compared with responses to the urban environment, they found lower mood disturbance, salivary cortisol in the green exposure environment, and favorable changes in heart rate variability indicators in the blue exposure environment. As such, many previous studies have revealed the potential of forest therapy to improve depression and anxiety.

However, although many previous studies have reported that forest-based activities are practical for physiological and physiological health, studies exploring the direct link between forest therapy and depression and anxiety are insufficient. In the previous three systematic literature reviews, it was reported that forest therapy was an effective intervention in improving depression and anxiety. However, because meta-analysis was not conducted, the effect size of forest therapy on depression and anxiety could not be analyzed [41,42,43]. In addition, Kotera et al. [44] systematically reviewed and meta-analyzed 20 studies. As a result, only six studies related to depression and five studies related to anxiety were RCT study designs. Since meta-analysis was performed on a small sample size, the effect size was likely overestimated or underestimated. Therefore, this review aims to systematically prepare evidence-based data by integrating forest therapy’s contents and effects based on previous studies related to depression and anxiety.

## 2. Materials and Methods

### 2.1. Literature Search

This systematic review was conducted following the Preferred Reporting Items for Systematic reviews and Meta-Analyses (PRISMA) guidelines [45] (Appendix B). We searched Scopus, PubMed, MEDLINE(EBSCO), Web of science, Embase, Korean Studies Information Service System, Research Information Sharing Service, and DBpia to identify relevant studies published from January 1990 to December 2020. The time frame was chosen because when we reviewed previous review papers [24,46,47,48] on the effects of forests on health, no literature was derived before 1990. So, we narrowed it from 1990 to 2020 in the search period. All search terms are listed in Appendix A, Table A1. The language of the published article was limited to English and Korean. Our review’s flow chart is shown in Figure 1.

### 2.2. Inclusion and Exclusion Criteria

This study is a systematic review and meta-analysis study to confirm the effect of forest therapy on anxiety and depression. The PICO-SD (Population, Intervention, Comparisons, Outcomes, Setting, Study design) framework was used to clarify the objectives of the review and facilitate the search strategy (Table 1). The main research questions of this review were the following: (1) how effective is forest therapy in improving depressive symptoms and anxiety? And (2) what quantity and quality of evidence is reported?

To be eligible for further analysis, primary studies need to (1) report an empirical intervention study, using pre-and post-intervention measures, (2) use mental health measures for depression or anxiety, (3) studies including at least one control group, and (4) been published either in English or Korean. Exclusion criteria were (1) review articles, (2) studies not including humans, (3) no interventions, (4) were case studies or qualitative studies, (5) no direct exposure in the forest environments, and (6) no results presented.

### 2.3. Data Extraction

Five authors (J.G.; M.S.; K.M.H.; G.M.M.; G.Y.) independently first screened the titles and abstracts of articles identified by the search strategy and then retrieved and screened the full-texts of these articles. After the full-text screening, the eight authors (J.G.; J.Y.; M.S.; K.M.H.; G.M.M.; G.Y.; S.H.; M.J.) extracted data from all studies that met our eligibility criteria. Data were extracted on author name and publication year, study design, participants characteristics, sample size, age of participants, intervention description and characteristics (i.e., type, duration, time, and frequency), control description, and effects on outcomes (Table 2). Any disagreements during screening or data extraction were resolved by another author (P.S.).

### 2.4. Risk-of-Bias (ROB) Assessment

Two authors (J.G. and J.Y.) independently assessed the ROB of RCTs, quasi-experimental studies using the revised Cochrane ROB tool for RCTs (ROB 2.0) [50], ROB in non-randomized studies of interventions (ROBINS-I) [51], respectively. Disagreements were discussed until a consensus was reached. The three ROB assessment tools used were composed of several categories: ROB 2.0 consists of five potential bias categories that are assessed as low ROB, some concerns, or high ROB utilizing a series of signaling questions. The categories are the randomization process, deviations from intended interventions, missing outcome data, measurement of the outcome, and selection of the reported results. ROBINS-I comprises seven bias categories: baseline confounding, selection of participants, classification of interventions, deviation from intended interventions, missing data, measurement of outcomes, and selection of reported results; each is evaluated as low, moderate, serious, or critical ROB or no information. Assessment of each category mentioned above provided the basis for an overall ROB judgment for the included studies.

### 2.5. Meta-Analyses

All statistical analyses were performed using the Comprehensive Meta-Analysis 3.3 (Biostat, Englewood, NJ, USA). As for the effect size, standardized mean difference (SMD) was selected as the analysis method to compare outcome variables with different measurement tools or measurement units, and Cohen’s d tends to overestimate the effect size when the sample is small, so Hedges’s g to correct it was calculated as the effect size. The magnitude of the effect size was defined as small (0.2 to under 0.5), medium (0.5 to 0.8), or large (above 0.8) [52]. Cochrane’s chi-square test was performed to verify the heterogeneity of the integrated effect size, and the I-square value was calculated with a significance level of less than 5%. The magnitude of heterogeneity was interpreted as follows: low (I^2^ = 0 to 24%), moderate (I^2^ = 25 to 49%), large (I^2^ = 50 to 74%), or extreme (I^2^ = 75 to 100%) heterogeneity [53].

This study confirmed heterogeneity between analysis studies, and a random-effects model was applied. The statistical meaning of effect size (d) was determined by the total effect test and 95% confidence interval (CI) and was based on the significance level of 5%. Subject characteristics, activity types, intervention time, and intervention frequency, which are parameters that can cause heterogeneity between individual studies, were set as modulating variables. In the meta-analysis, heterogeneity was confirmed and adjusted using the CMA 3.3 program, meta-ANOVA was performed in categorical cases according to the attributes of the adjustment variable, and meta-regression was applied to analyze the moderating effect.

Publication bias was assessed using funnel plots and Egger’s regression asymmetry test, used only when at least ten studies were included in a meta-analysis [53]. We considered *p* < 0.05 in asymmetrical funnel plots to indicate potential publication bias.

## 3. Results

### 3.1. Search Results

Figure 2 summarizes the selection process for the 20 studies in this review. After initially identifying 164,587 records (Scopus, *n* = 20,711; PubMed, *n*= 5777; MEDLINE, *n* = 352; Web of science, *n*= 14,138; Embase, *n*= 853; RISS, *n*= 5277; DBpia, *n* = 117,479). The full text of 13,698 potential studies was then screened and extracted for further details. We removed 4379 duplicates and then reviewed titles and abstracts of 9319 studies. We excluded 9299 publications for the following reasons: (1) 9211 records not based on forest therapy-related intervention evaluating depression or anxiety outcomes, (2) 62 records not RCTs study design or quasi-experiments, such as cross-over design, one group pretest-posttest design, and qualitative studies, (3) 14 records were indirectly exposed in the forest, such as Virtual reality (VR) and 2D-image, and (4) 12 records did not provide results data.

We finally included five RCTs [54,55,56,57,58] and 15 quasi-experiments [59,60,61,62,63,64,65,66,67,68,69,70,71,72,73] in the qualitative analysis and meta-analysis.

### 3.2. Characteristics of Selected Studies

Table 2 summarizes the characteristics of the 20 selected studies conducted in the Republic of Korea except one conducted in Poland [54]. In the publication year, three studies (15.0%) were published from 2011, eight studies (40.0%) from 2012 to 2015, and nine studies (45.0%) from 2016 to 2019. Regarding the study design, five studies used a randomized controlled trial (RCT) design, and 15 studies used a quasi-experiment design. Participants were characterized by nine studies (45.0%) of healthy people, including college students, office workers, middle-aged women, and the elderly, followed by four studies (20.0%) of chronic diseases such as cancer, cerebral infarction, and mild cognitive impairment, as well as infants and adolescents. Moreover, there are three studies (15.0%) on patients with mental illness. The sample size ranged from 20 [61] to 240 [71], and in almost all the studies, the sample size was from 51 to 100.

#### 3.2.1. Format and Content of Forest Therapy

Table 2 summarized the 20 selected studies, shown in the participant’s characteristics, intervention description and characteristics (i.e., type, duration, time, and frequency), and effects on outcomes. The 20 studies varied in terms of the format and content of the forest therapy. The length of duration that the intervention varied from one day to 7 months. Among them, six studies provided one-time intervention, one was one-day intervention [54], three-day intervention [65], one-day two-night intervention [62], two-day three-night intervention [71], three-day four-night intervention [57], and nine-day intervention [58], respectively. In the other 14 studies, the duration of intervention ranged from 2 weeks to 7 months. As for the frequency of intervention, eight studies were the most common once a week, followed by two times a week and three times a week, and one study did not report the frequency of the intervention [69]. The intervention time was 40 min to 120 min, which is the nine studies provided within 2 h and one study conducted to 3 h and 5 h, respectively. Three studies did not report duration details of the intervention [64,69,72].

Regarding the content of forest therapy, walking in the forest was the key component of forest therapy. Other therapeutic activities included in forest therapy were forest viewing, meditation, mindfulness-based cognitive behavior therapy, bodily stimulation exercise, and recreation (e.g., bingo game in the forest, treasure hunt in the forest, touching a natural object). On the other hand, the most common type of control intervention was “normal daily routines (12 out of 20 studies)”. Otherwise, a general program was performed, or the same activities were performed in other places such as cities and indoors rather than forests.

#### 3.2.2. Depression or Anxiety Measures

As for the measurement used as self-reported for depressive symptoms, Beck Depression Inventory (BDI), which is widely used in adults, was the most used with eight studies [56,57,58,60,62,64,70,73]. Next, three studies [54,64,65] used Profile of Mood States, and two studies used Hamilton Rating Scale for Depression (HRDS) [57,70], Children’s Depression Inventory (CDI) [59,71], and Geriatric Depression Scale Short Form (GDSSF) [55,68], respectively. Other scales used to measure depression were the Montgomery-Asberg Depression Rating Scale (MADRS) [70], Hospital Anxiety and Depression Scale (HADS) [65], Center for Epidemiologic Studies Depression Scale (CES-D) [72], Korean Depression Scale (KDS) [63], Zung Self-Rating Depression Scale (ZSDS) [66], and Stress Response Inventory (SRI) [61].

The most used self-reported measurements for anxiety were the State-Trait Anxiety Inventory (STAI) [57,67,73] and Profile of Mood States (POMS) with three studies, respectively. In addition, the Hospital Anxiety and Depression Scale (HADS) and Korean Preschool Daily Stress Scale (KPDSS) [69] were used. Detailed information on the measurement tools included in these studies is summarized in Table 2.

**Table 2 ijerph-18-12685-t002:** Characteristics of the 20 included studies.

Author (Year)	Country	Study Design	Participants	Sample Size (Intervention/Control)	Intervention	Control	Duration (Time/Frequency)	Outcome
Bielinis et al. (2019) [54]	Poland	RCT (2 group)	Grauduate students—nonforestry course (20.97 ± 0.65 y)	32 (16/16)	Forest recreation—relaxation: standing and viewing	Urban street environment at the urban point	1 day (15 min/NA)	POMS: T-A *#↓ D *#
Jun et al. (2019) [55]	Republic of Korea	RCT (2 group)	Mild cognitive impairments (65~100 y)	57 (28/29)	Forest therapy program—getting closer to the forest: making a natural objects, observing the habitats of animal, etc.,	Daily routine activities	8 sessions (120 min/Once a week)	GDSSF *#↓
Bang et al. (2016) [56]	Republic of Korea	RCT (2 group)	University staffs (42.22 ± 11.44/37.37 ± 9.32 y)	45 (18/27)	Walking in the urban forest during lunch time	Daily routine activities	10 sessions (40 min/Twice a week)	BDI: N/A
Chun et al. (2016) [57]	Republic of Korea	RCT (2 group)	Chronic alcoholics (45.26 ± 3.89 y)	92 (47/45)	(1) Forest therapy program—forest walking, experiencing the forest through all five senses, meditation, etc., (2) Staying at a recreational forest site	Staying in an urban hotel	3 Night 4 Days	(1) STAI #↓ (2) BDI *↓ (3) HRSD *↓
Shin et al. (2012) [58]	Republic of Korea	RCT (2 group)	Chronic alcoholics (45.26 ± 3.89 y)	92 (47/45)	Forest therapy camp −Interacting with nature/forest, challenge, self-introspection, etc.,	Daily routine activities	9 days	BDI #↓
Bang et al. (2018) [59]	Republic of Korea	Quasi-experimental design (2 group)	Elementary students (11~13 y)	52 (24/28)	Health Promotion Program using urban forest—five senses experience in urban forest, forest walking/exercise, and playing with natural materials etc.,	Routine programs (e.g., supplementary learning such as math or English, reading, art)	10 sessions (120 min/Once a week)	CDI *#↓
Bang et al. (2017) [60]	Republic of Korea	Quasi-experimental design (2 group)	University students & Graduate students (24.3 ± 4.19 y)	99 (51/48)	Campus forest-walking program	Daily routine activities	6 sessions (60 min/Once a week)	BDI *#↓
Choi et al. (2016) [61]	Republic of Korea	Quasi-experimental design (2 group)	Middle aged woman (53.9 ± 2.69/55.5 ± 1.84 y)	20 (11/9)	Forest walking	Treadmill walking	36 sessions (80 min/Three times a week)	SRI: Depression #↓
Han et al. (2016) [62]	Republic of Korea	Quasi-experimental design (2 group)	Full-time employees (41.6 ± 6.5/37.5 ± 8.4 y)	61 (33/28)	Forest therapy program− forest walking, music therapy, stimaltion bodily exercise, mindfulness-based meditation, herbal tea time, etc.,	Weekend routines except visiting natural environments	One night two days	BDI #↓
Oh (2016) [63]	Republic of Korea	Quasi-experimental design (3 group)	Middle aged woman (40~56 y)	60 (20/20/20)	(1) Forest therapy program—exercise, relaxation, diet, etc., (2) Urban forest therapy	Daily routine activities	4 sessions (300 min/Once a week)	KDS *#↓
Kim, M et al. (2015) [64]	Republic of Korea	Quasi-experimental design (2 group)	Psychiatric inpatients (35~56 y)	20 (10/10)	Forest therapy program−handkerchief dyeing, decorating the frame with natural object, group work, etc.,	Daily routine activities	10 sessions (NA/5 times a week)	(1) POMS: T-A: N/A D: N/A (2) BDI: N/A
Kim, Y et al. (2015) [65]	Republic of Korea	Quasi-experimental design (2 group)	Cancer patients (30~79 y)	53 (37/26)	Forest therapy program−wildflow exploration, playing in the forest, mindfulness, flower tea therapy, sharing feelings, etc.,	Daily routine activities	3 days (120 min)	(1) HADS #↓ Depresion #↓ Anxiety #↓ (2) POMS: T #↓ D #↓
Choi et al. (2014) [66]	Republic of Korea	Quasi-experimental design (2 group)	Cancer patients (more than 50 y)	53 (26/27)	Forest therapy program−forest walking and abdominal breathing, touching wood, meditation, cooperativity activities, etc.,	Daily routine activities	8 sessions (120 min/Once a week)	ZSDS *#↓
Kim and Lee (2014) [67]	Republic of Korea	Quasi-experimental design (2 group)	University students	67 (35/32)	Forest therapy program− greeting with objects in forest, meditation, forest walking, mandara made from rope, my look in forest, etc.,	Daily routine activities	8 sessions (50 min/Twice a week)	STAI #↓
Lim et al. (2014) [68]	Republic of Korea	Quasi-experimental design (3 group)	Elderly people (50~99 y)	64 (22/21/21)	(1) Forest therapy program− being familiar with forest, activating sense of nature, feeling happyness in forest, etc., (2) Forest therapy in indoor	Daily routine activities	11 sessions (90 min/Once a week)	GDSSF *#↓
Shin (2012) [69]	Republic of Korea	Quasi-experimental design (2 group)	Infants (71~73 month)	63 (25/38)	Forest Kindergarten Education Programforest walking, nature observation activities, drawing and exploring in the forest, etc.,	Basic curriculum	7 months	KPDSS: Anxiety #↓
Woo et al. (2012) [70]	Republic of Korea	Quasi-experimental design (4 group)	Major depressive disorders (43.39 ± 12.14/44.26 ± 13.49/48.40 ± 15.00/48.79 ± 9.63 y)	81 (28/21/15/17)	(1) Forest therapy program−Mindfulness-based cognitive behavior therapy, forest activities, meditation, promoting interpersonal relationships, etc., (2) Hospital program (3) Forest bathing	Conducted same activities in the indoor	4 sessions (180 min/Once a week)	(1) HRSD #↓ (2) BDI: N/A (3) MADRS #↓
Cho et al. (2011) [71]	Republic of Korea	Quasi-experimental design (2 group)	Childrens (8~12 y)	240 (120/120)	Forest experience camp—forest festival, forest walking, field day in the forest, forest mission impossible, and playing in the forest, etc.,	Daily routine activities	2 Night 3 Days	CDI *↓
Kim (2011) [72]	Republic of Korea	Quasi-experimental design (2 group)	Middle school students (14~16 y)	80 (40/40)	Forest therapy program− create an ecological map, talking to nature, to bring out the sense of nature, etc.,	General class	10 sessions (NA/Once a week)	CES-D *↓
Song (2009) [73]	Republic of Korea	Quasi-experimental design (2 group)	Unmarried mothers (10 s~30 s)	75 (35/35)	Forest therapy program− forest meditation, reviving the dull senses, prenatal care in the forest, etc.,	Daily routine activities	24 sessions (120 min/Twice a week)	(1) BDI *↓ (2) STAI *↓

Notes: RCT, Randomized controlled trial; BDI, Beck Depression Inventory; POMS, Profile of Mood States; T–A; tension-anxiety; D, depression; HRDS, Hamilton Rating Scale for Depression; CDI, Children’s Depression Inventory; GDSSF, Geriatric Depression Scale Short Form; MADRS, Montgomery-Asberg Depression Rating Scale; HADS, Hospital Anxiety and Depression Scale; CES-D, Center for Epidemiologic Studies Depression Scale; KDS, Korean Depression Scale; ZSDS, Zung Self-Rating Depression Scale; SRI, Stress Response Inventory; STAI, State-Trait Anxiety Inventory; HADS, Hospital Anxiety and Depression Scale; KPDSS, Korean Preschool Daily Stress Scale; y, years old; * Significant inra-group differences; # Significant inter-group differences; “↓”, indicators decline; N/A, no report. Underlined studies were written in Korean.

### 3.3. Quality Assessment

Appendix A, Table A2 shows the ROB assessments for the 20 included studies. Of the 20 studies included, five studies were RCTs, so ROB 2.0 tools were used, and 15 studies were conducted for risk of bias using ROBINS-I tools, a non-randomized controlled experimental evaluation tool. The overall quality of five RCTs raised potential concerns in three [54,55,57] and high in two [56,58]. The risk of bias in the randomization process was high risk in two [56,58], some concerns in two [55,57], and low risk in one [54]. Although all RCTs were rated as low ROB for Deviation from intended interventions, missing outcome data, and selection of the reported results, all RCTs raised some concerns about the measure of the outcome data.

We assessed overall ROBs as high for nine quasi-experimental studies [63,65,66,67,68,69,71,72,73] and moderate for six studies [61,62,63,64,70] due to the blinding of outcome assessors and potential sources of knowledge of the intervention received in bias arising from the measurement of outcomes. However, by item, it was judged that the risk of bias was all low ROB for baseline confounding, selection of participants, classification of intervention, deviation from intended interventions, missing data, and selection of reported results.

### 3.4. Effects of Forest Therapy on Depression

Forest plots in Figure 3 display results of the meta-analyses for the effects of forest therapy on depression. Of the 20 studies analyzed in this review, 18 studies confirmed depression as a dependent variable. In 18 studies, the effect size for 23 result values was calculated, and the studies were heterogeneous (I^2^ = 89.35%). Therefore, the effect size was calculated with a random effect model. The overall effect size was 1.133 (95% CI: −1.491 to −0.775, *p* < 0.0001), indicating a high effect size.

### 3.5. Effects of Forest Therapy on Anxiety

Forest plots in Figure 4 display results of the meta-analyses for the effects of forest therapy on anxiety. Of the 20 studies analyzed in this review, eight studies confirmed anxiety as a dependent variable. The effect size for nine result values was calculated in eight studies, and the studies were heterogeneous (I^2^ = 93.35%). Therefore, the effect size was calculated with a random effect model. The overall effect size was 1.715 (95% CI: −2.519 to −0.912, *p* < 0.0001), indicating a high effect size.

### 3.6. Verification of Heterogeneity of Effect Size: Analysis of Modulating Effect

#### 3.6.1. Depression

To analyze the cause of heterogeneity in effect size, meta-ANOVA was conducted using the sub-groups of this review. Table 3 shows subgroup analyses by participant characteristics, activity type (day and session types), activity contents, intervention time, and duration as modulating variables.

In subgroup analyses, the intervention time was significant modulating variables of effect size. As a result of comparing the effect sizes by intervention time, 121 min or more (Hedges’s g = 2.035; 95% CI: −2.790 to −1.279) and 61 to 120 min (Hedges’s g = 0.862; 95% CI: −1.396 to −0.327) had a large effect size in the depression. However, 60 min or less (Hedges’s g = 0.356; 95% CI: −1.089 to 0.376) had a small effect size, which is no significant difference.

Compared with participant’s characteristics, mental disorder (Hedges’s g = 1.522; 95% CI: −2.190 to −0.854) was the highest, followed by healthy adults (Hedges’s g = 1.242; 95% CI: −1.783 to −0.701), patients with chronic diseases (Hedges’s g = 1.009; 95% CI: −1.974 to −0.043) had a large effect size in the depression. However, children and adolescents had a small effect size. However, there was no statistically significant difference.

By activity content, forest therapy programs (Hedges’s g = 1.291; 95% CI: −1.690 to −0.892, *p* < 0.001) had a larger effect size of improving depression than forest walking (Hedges’s g = 0.252; 95% CI: −1.249 to 0.745, *p* = 0.620) but showed no significant difference (Q = 3.595, df = 1, *p* = 0.058). In addition, there was no significant difference between subgroups in activity type and duration for effect size.

#### 3.6.2. Anxiety

To analyze the cause of heterogeneity in effect size, meta-ANOVA was conducted using the sub-groups of this review. Table 4 shows subgroup analyses by participant characteristics, activity type (day and session types), intervention time, and duration as modulating variables.

In subgroup analyses, the activity type was the significant modulating variable of effect size. Day type (Hedges’s g = 2.711; 95% CI: −2.573 to −0.883) had a significantly larger effect size of improving anxiety than session type (Hedges’s g = 0.990; 95% CI: (−2.108 to 0.129).

By participant’s characteristics, the people with chronic disease (Hedges’s g = 3.236; 95% CI: −4.892 to −1.580) had a larger effect size of improving anxiety than healthy people (Hedges’s g = 1.442; 95% CI: −2.832 to −0.051) but showed no significant difference. In addition, there was no significant difference between subgroups in the intervention time and duration for effect size.

### 3.7. Publication Bias

The meta-analyses for the effect of forest therapy on the depression and anxiety domains showed visual evidence in asymmetric funnel plots (Appendix A, Figure A1) and significance on Egger’s regression asymmetry tests (t = 3.25, *p* = 0.004 for depression and t = 2.40, *p* = 0.047 for anxiety) (Appendix A, Table A3).

On the other hand, the impact of automatic missing data was analyzed when the asymmetric funnel plot was symmetrically changed using Trim-and-Fill proposed by Duval and Tweedie [74]. There was a publication bias in the overall effect size of this review. As a result, it was found that the effect size automatically corrected in consideration of the possibility of publication bias was significant as 1.342 (95% CI: −1.73 to −0.95) for depression and 2.476 (95% CI: −3.50 to −1.46) for anxiety. Therefore, it was confirmed that the overall effect size on depression and anxiety was not affected by publication bias.

## 4. Discussion

### 4.1. Summary of Findings

This review attempted to confirm the effect of forest therapy by systematically examining and meta-analyzing the effects of forest therapy on depression and anxiety over the past 30 years. Our findings have shown that forest therapy has large effect sizes, not just significant effect evidence on depression and anxiety. These results are consistent with the previous results on psychological effects, including depression and anxiety [44,75,76]. For example, a meta-analysis of the effect of improving depression of forest therapy in 13 studies by Rosa et al. [75] showed that forest therapy is a more effective short-term intervention for adult depression prevention and treatment with an average effect size of 1.18 (95% CI: 0.86 to 1.50), *p* < 0.0001). Compared with no intervention/treatment, this study reported that participants in the forest healing group were 17 times more likely to achieve depression relief (Risk Ratio = 17.02, 95 % CI [3.40, 85.21], *p* = 0.0006), and 3 times more likely to decrease by 50% for depressive symptoms (Risk Ratio = 3.18, 95 % CI [1.94, 5.21], *p* < 0.00001). In addition, Kotera et al. [44] showed an effect size of −2.54 MD (95% CI: 3.56–1.52) for depressive symptoms in six RCT studies and a large effect size of −8.81 MD (95% CI: −21.91–3.57) for anxiety symptoms in five RCT studies.

The results of this review were analyzed in subgroups based on participant characteristics, activity type, intervention time, duration, and intervention content due to extreme heterogeneity between included studies. As a result, we found significant differences between participant characteristics and activity type in the sub-group.

In terms of participant characteristics, the effect size of people with mental health conditions (Hedges’s g = 1.522) was the largest, followed by healthy adults (Hedges’s g = 1.242), and chronic disease patients (Hedges’s g = 1.009), showing large effect sizes for depression. However, children and adolescents (Hedges’s g = 0.136) had a small effect size. These findings suggest that forest environments can significantly lower depression in adults, especially people with a mental health condition, more than in children or adolescents. The findings are partly consistent with the results of previous studies dealing with the possibility of forest therapy to treat specific mental and physical conditions, such as depression [77], high-risk stress groups [35], hypertension treatment [78], and patients with severe exhaustion disorder [79]. In particular, Furuyashiki et al. [77] investigated forest therapy’s physiological and psychological effects on workers. They proved that forest therapy has a significantly positive effect on mental health compared to those who do not take part. Therefore, it will be necessary to actively use forest therapy to improve the psychological health of chronically ill or mentally ill patients.

In the case of activity type, we found that the effect of forest therapy on anxiety was statistically significantly higher in the day-type forest therapy than in the session-type forest therapy. In a meta-analysis study on the effectiveness of previous forest therapy programs, the effect size was more significant in the session-type activity type than in the day-type [80]. In addition, Christup [81] argued that the participants are more likely to change when intervening in the long term than in the short term. The results of this review were different from those of previous studies [80,81]. However, it is thought that the previous study [80] may differ from the results of this study as the effect size of meta-analysis that integrates depression and anxiety as well as other psychological functions, cognitive functions, social functions, and physiological functions. In addition, studies that directly compare the effects of short-term forest therapy, such as one night and two days and long-term forest therapy conducted more than once a week with 12 sessions a week, are insufficient. Therefore, it is thought that future research is needed on the difference in the effectiveness of each activity type on improving anxiety.

In addition, there was no significant difference when the subgroup analysis was conducted on the intervention contents. However, the forest therapy program showed a more significant effect than simply forest walking activities. The results of this study are partly consistent with the results of previous studies that show that forest therapy programs were more effective in psychological recovery than forest walking activities [82,83]. Thus, it may be more effective in improving depression and anxiety than walking in the forest.

In the case of intervention time, we found that the longer the intervention time, the greater the effect of forest therapy on depression and anxiety. In particular, the effect size for depression was largest when the intervention time was more than 120 min, followed by more than 61 min and less than 120 min and less than 60 min. The results of this study are partly consistent with the results of previous studies that when the intervention time is three hours, the effect size is larger than within an hour, 1 h to two hours, and 2 h [84]. According to the previous study, it was reported that organizing the intervention time within an hour has limitations in operating in-depth programs [85] and helps improve the effectiveness of activities by securing sufficient intervention time [86]. Therefore, this result shows that spending enough time in the forest is more effective in improving depression and anxiety.

The results of this study are supported by theories used as the basis for explaining natural recovery in environmental psychology. The most influential frameworks in explaining the effects include Kaplan’s [87,88] attention restoration theory (ART) and Ulrich’s [89] stress reduction theory (SRT). According to the attention restoration theory (ART) [87,88], exposure to nature such as forests can reduce mental fatigue or psychological stress and restore attention to a more positive emotional and psychological response. The brain’s capacity to focus on a specific stimulus or task is limited, resulting in ‘directed attention fatigue.’ ART proposes that exposure to natural environments encourages more effortless brain function, thereby recovering and replenishing its directed attention capacity. Several studies support this theory, showing that staying in natural environments positively affects recovery from directed attention fatigue [90,91]. The stress reduction theory (SRT) [89], another commonly used theory in environmental restorativeness, focuses on psychophysiological stress. The SRT suggests that the natural environment affects the emotional state by promoting stress recovery, evoking positive emotions, and blocking negative emotions through psychophysiological pathways [92,93]. It has been reported that the evidence found to date is that viewing or visiting the natural environment can reduce blood pressure and stress hormone levels and positively affect mood states [94,95]. In addition to theories, Hartig’s in press [96] relational restoration theory (RRT) and Collective restoration theory (CRT) are more recent approaches to restoration [97]. The theories are based on the argument that restoration does not occur in a social vacuum. RRT suggests that small groups of social resources can be reduced. These resources can be restored, for example, by spending time together in nature. Likewise, CRT suggests that depletion and restoration of social resources can be collective. For example, the summer vacation period for a specific population may be correlated with an increase in group welfare during leisure. Conversely, cold summer weather can constrain group recovery activities outdoors and negatively affect downstream groups such as by increasing stress. In addition to these two theories, vitality, the feeling of activation in a recovery environment such as a forest, is defined as “having physical and mental energy” and is related to many inactive positive emotions such as satisfaction and happiness [98,99]. Restoration refers to the process of renewing, restoring, or rebuilding reduced physical, psychological, and social resources or functions in continuous efforts to meet adaptation needs [100]. Therefore, the natural restoration environment itself as a forest may be essential in improving anxiety and depression.

In addition, five senses stimulation due to forest characteristics such as forest scenery, forest sound, scent, and various tactile elements may be a mechanism factor related to improving anxiety and depression. It has been proven in many studies that forest landscapes promote mental and physical relaxation and improve stress resilience [24,99]. For example, Song et al. [39] reported that viewing at the forest landscape significantly reduced depression, tension-anxiety, and anxiety rather than the city. As a follow-up study, Song et al. [101] showed that changes in depression after viewing the forest landscape showed a significant correlation with state anxiety. This correlation showed a greater difference in individuals with high anxiety.

Auditory stimulation by the forest, such as the sound of leaves shaking in the wind, singing birds, and the sound of flowing streams, can contribute to psychophysical relaxation and stress recovery [102,103,104]. For example, Zhang et al. [103] reported that the acoustic and visual comfort given in the green environment had a strong positive correlation with low depression and anxiety. In particular, acoustic comfort showed a more significant influence than visual comfort. In addition, Ochiai et al. [104] reported that as a result of a survey of the effects of forest sound on gambling addicts, forest sound significantly reduced depression and tension-anxiety rather than urban sound. Thus, it has been proven that visual and auditory stimuli in the forest can significantly improve emotional depression and anxiety.

As another critical factor, trees in the forest release biogenic volatile organic compounds (BVOCs) such as limonene, alpha-pinene, and beta-pinene, which affects not only human health in terms of anti-inflammatory, antioxidant, or neuroprotection activities [105,106,107], but also benefits psychological and cognitive processes [108]. For example, Lee et al. [109] investigated the effect of cypress-oriented inhalation on stress and depression in college students. As a result, participants who inhaled cypress orientation had significantly decreased depression than in the pre-test, while those who did not inhale orientation had increased depression compared to the pre-test. In addition, although it is a preclinical study of animal behavior models, BVOCs showed anti-anxiety [110,111] and antidepressant properties [112,113]. It has been proven that tactile stimulation caused by the contact of bark and plant leaves of trees can also relax and stimulate parasympathetic nerve activity more than touching other materials [114,115]. Overall, tactile stimulation caused by touching living forest plants can play a role in calming effects. However, the primary studies on forest-based intervention did not elaborate on the forest structure where the intervention was performed. In addition, it is unclear what kind of forests are most effective in improving depression and anxiety. Future research will require research on what forest structure (e.g., the intensity of thinning, biodiversity, tree density, tree species, etc.) can maximize mental health effects.

These findings also support the evidence that forest therapy, which is part of nature therapy, can be used as a non-pharmacological intervention to improve depression and anxiety. Mental health problems are becoming more and more severe as many people are excessively exposed to stress due to urbanized society [116,117]. For example, according to Sundquist et al. [118], both women and men reported that urban residents had a 20% higher risk of depression than rural residents. The higher risk of anxiety disorders was also about 21% [119]. In addition, McKenzie et al. [117] reported that urban environments are associated with higher prescription rates of antipsychotic drugs for anxiety, depression, and mental illness. However, some studies have reported no difference in the proportion of residents’ depression and anxiety symptoms between suburban and rural areas [120], but instead causes more mental problems in the countryside [121]. This shows that not only green space in the area where residents live but also variables related to demographic factors such as age, gender, ethnicity, and cultural factors affect mental health. Therefore, it is necessary to provide more insight at the international level through research that considers aspects of other cultural realities.

According to the World Health Organization [122], more than half of the world’s population lived in urban environments in 2014, increasing to 65% by 2030. Accordingly, medical costs are also incurred, so reducing medical costs at the preventive level is a socially important issue. According to Buckley et al. [123], the economic value of nature reserves is evaluated based on the mental health of visitors, accounting for about 8% (about 6 trillion dollars) of the world’s gross domestic product. Becker et al. [124] also estimated the association between the five vegetation ratios of forests, shrubs, grasslands, agriculture, and urban vegetation and medical insurance premiums for 3086 counties in the United States. As a result, the ratio of forests and shrubs was significant and negatively correlated with medical insurance expenditure. In other words, it was found that 1% of the land in a country covered with forests was associated with low health insurance spending of $4.32 per person per year. Thus, a wide range of applications of forest therapy can improve the country’s overall health and lead to significant savings and productivity benefits in terms of health care and welfare systems.

The psychological benefits of forests are significant. As a preventive dimension, the forest environment and the urban green space can play an essential role in promoting mental health [101]. This is because urban green spaces such as urban forests provide cost-effective, simple, and accessible methods to improve individual quality of life and health in urban areas. Therefore, it is vital to prepare a way for urban residents to improve and access their daily lives by utilizing the restorative environment of the forest. To do this, urban planners need to pay more attention to the maintenance and increase in accessible green spaces in urban areas.

### 4.2. Limitations

The limitations of this study are as follows. First, the magnitude of the overall heterogeneity among studies was considerable. Accordingly, although we used a random effect model to determine the effect size, the primary studies’ interpretation may be limited due to high heterogeneity. Furthermore, we included not only five RCTs but also 15 quasi-RCTs. NRCTs tend to have a higher risk of bias than RCTs due to confounding because participants’ allocation to intervention may be related to baseline variables affecting outcomes. So, it leads to methodological heterogeneity.

Second, most primary studies showed a moderate or high risk of bias. It was almost impossible to apply blind forest therapy interventions that did not provide interventions to participants, therapists, and assessors. In addition, questionnaires measuring depression and anxiety have been self-reported. Random errors caused by incorrect memories could reduce the size of the link between forest therapy and depression and anxiety. To reduce the random error, many studies will need an objective measure for assessing the level of depression or anxiety. This is because significant correlations between physiological findings such as EEG asymmetry [125,126], heart rate variability [127,128], and perceived levels of depression and anxiety have been reported. Therefore, future studies should require better-designed, controlled intervention studies and reliable physiological measures in addition to self-reported questionnaires to find the effects of forest therapy.

Third, the effects achieved by forest therapy were usually compared to control groups without any specific intervention, and controls merely followed their “daily routine” [55,56,58,63,64,65,66,67,68,71,73]. These studies investigated the effects of depression and anxiety of forest therapy intervention without explicitly explaining the contribution of the forest environment to the achieved effects. It is still unclear whether the same results were achieved compared to the same intervention in an environment outside the forest. This point shows that special attention is paid to the appropriate control group when selecting a study design to prove the effectiveness of forest therapy.

Fourth, primary studies were carried out in Korea except for one study [49]. Therefore, we should be careful when interpreting the results, and the need for broader geographic application is emphasized in forest therapy effects on depression and anxiety.

Fifth, in the process of systematic review, it is possible that unpublished studies or studies published in other languages were excluded because only studies published in Korean and English over the past 30 years were searched. Although we could not conduct literature searches in Japanese or Chinese databases conducting much research on forest therapy, we used four major international and two major Korean databases. It seems that our approach sufficiently identified many for this systematic review and meta-analysis.

Sixth, care should be taken in interpretation, as the effect size may be overestimated or underestimated by a meta-analysis of 20 primary studies. Therefore, based on this review, we believe that repeated studies on meta-analysis are necessary.

Seventh, the selected studies did not conduct follow-up evaluation. The absence of follow-up evaluation may impair the effectiveness of clinical studies [129], and this is because it is unclear whether the effect of forest therapy can continue. Therefore, future studies need follow-up evaluations to evaluate the long-term effects of improving depression and anxiety in forest therapy.

Eighth, we could not conduct a pre-register this study in the protocol registration. Future studies should need to register protocol before starting the meta-analysis, specifically before starting the data extraction. This registration will also ensure a clear documented research plan prior to commencing the systematic review. It will allow the researchers to understand the questions already registered or reported by other scientific investigators.

Despite these limitations, this study provided an understanding of the therapeutic benefits of forest therapy. Therefore, we believe that forest therapy should be actively used as preventive management and non-pharmacologic treatment for improving depression and anxiety. This is in line with the growing support for therapeutic activities in contact with nature as a non-pharmacologic intervention method for preventing and treating mental health problems.

### 4.3. Future Research

The directions for further development of this field are as follows. First, our findings emphasize the need for methodologically stricter RCTs to investigate the effects of forest therapy on depression and anxiety. Future research needs to improve the methodological quality of the studies while reducing the risk of bias in the study to increase evidence-based reliability. The most common concern in previous studies is the lack of blinding of participants, therapists, and evaluators who contribute to measurement results and interventions. In other words, many studies have not performed randomization and blinding. This means that the expectations of participants, therapists, and evaluators can affect the research results. If it is revealed that it is a group allocation, participants interested in forest therapy may become more vulnerable to the placebo effect. In addition, if blinding of researchers and therapists is not performed, this can also lead to placebo effects of participants. Therefore, future studies should thoroughly conduct randomization and blinding to conduct RCT studies.

Second, it is necessary to observe the continuous effects of forest therapy for a long time. After the intervention, the results of several follow-up measures (e.g., three months, six months, 12 months) should be evaluated. In addition, this includes various exposure times (e.g., 30 versus 60 versus 90 min), frequency of exposure (e.g., weekly versus every four weeks), that is, the optimal duration for forest therapy to perform the best effect.

Third, it is necessary to investigate whether a specific forest structure can have the best effect on improving depression and anxiety. For example, it will be essential to study the recovery effect according to the stand structure (e.g., forest type, tree density, distribution of canopy layer) and whether it is managed (natural forests vs. managed forests). In addition, seasonal changes in forests can have a significant impact on the recovery effect.

Fourth, it is necessary to research which activities and combinations of forest therapy programs effectively improve depression and anxiety. The activities applied to forest therapy programs are very diverse. Key elements are exposure to forest environments with all five senses (visual, smell, hearing, tactile, taste), which can be combined with various recreational activities and cognitive behavioral therapy as well as meditation, walking, and exercise. If it consists of customized activities to improve depression and anxiety, it will be possible to maximize the health benefits of forest therapy programs.

Fifth, researchers need to describe research methods and results in detail to derive more reliable meta-analysis results in future multidisciplinary studies, including environment and public health. Some studies cannot be included in the meta-analysis due to the unavailable results (e.g., missing outcomes, unrecorded exposure levels, research design methods).

Sixth, it is necessary to evaluate studies using tools suitable and validated for individual studies’ research design. Most individual studies’ quality evaluation is conducted in the case of meta-analysis studies published on the environment and public health. However, careful consideration of quality evaluation tools is required since the intervention effect can be inflated or reduced if the quality of individual studies is not correctly evaluated [130].

## 5. Conclusions

This systematic review and meta-analyses summarized the currently available evidence on the association between forest therapy and depression and anxiety. As a result, it demonstrated that forest therapy has a large effect on alleviating depression and anxiety. Therefore, it is necessary to actively apply forest therapy to improve depression and anxiety in the future.

## Figures and Tables

**Figure 1 ijerph-18-12685-f001:**
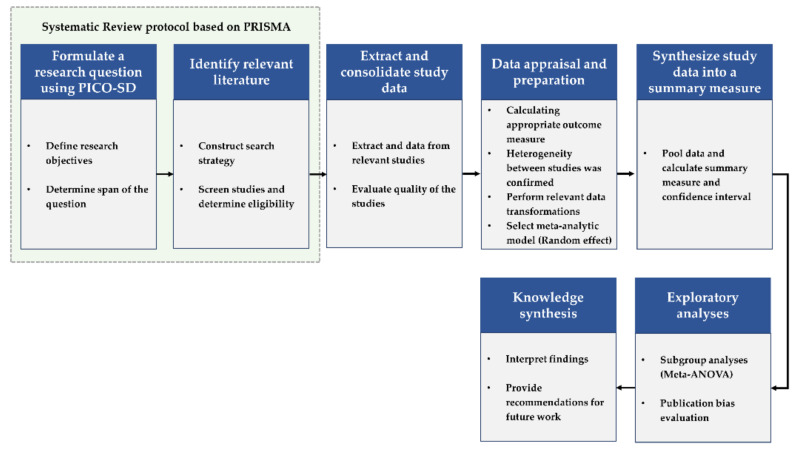
Systematic Review and Meta-analysis process diagram (Adapted from ref. [49]).

**Figure 2 ijerph-18-12685-f002:**
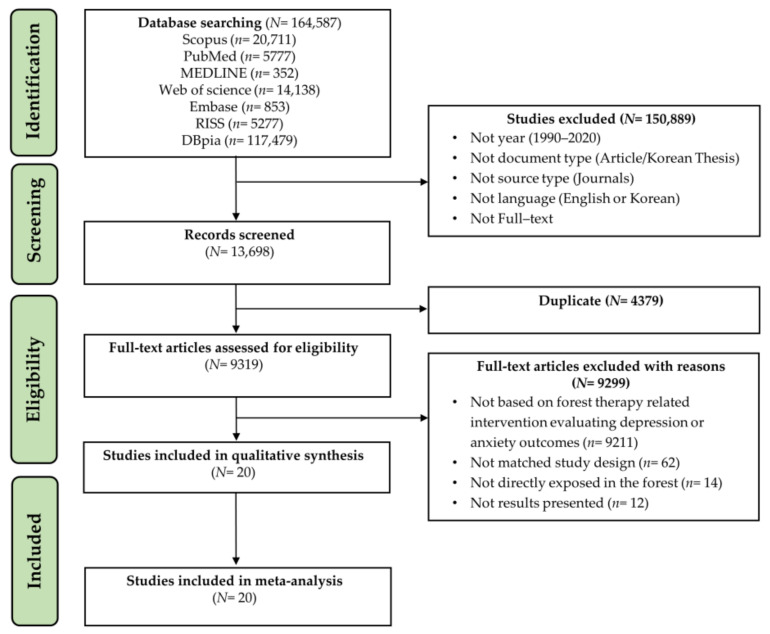
PRISMA flow chart of study selection.

**Figure 3 ijerph-18-12685-f003:**
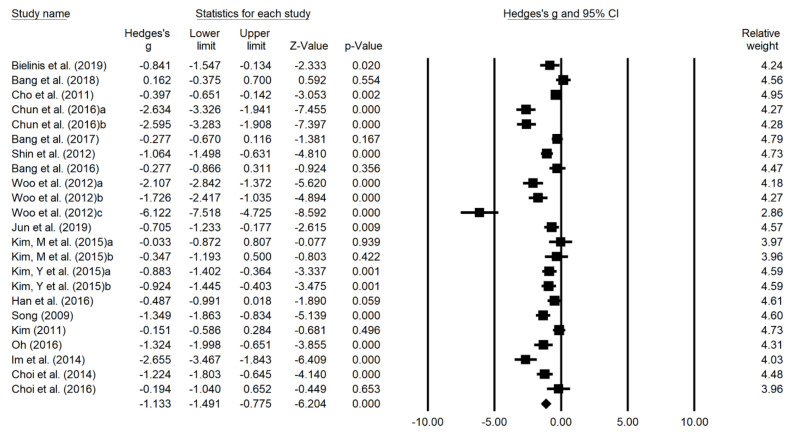
Forest plot the effects of forest therapy on depression. Heterogeneity: Tau^2^ = 0.656; Q = 206.490; I^2^ = 89.35%; Test overall effect Z = −6.204 (*p* < 0.001).

**Figure 4 ijerph-18-12685-f004:**
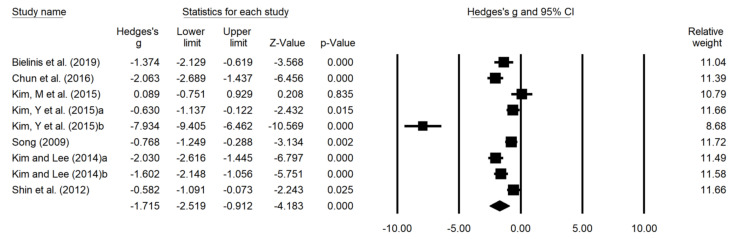
Forest plot the effects of forest therapy on anxiety. Heterogeneity: Tau^2^ = 1.374; Q = 120.259; I^2^ = 93.35%; Test overall effect Z = −4.183 (*p* < 0.001).

**Table 1 ijerph-18-12685-t001:** Inclusion criteria based on PICO-SD (Population, Intervention, Comparison, Outcomes, Setting, and Study design).

PICO-SD	Inclusion Criteria
Population	Studies with humans, healthy or not.
Intervention	Studies must include experimental conditions in which participants were directly exposed to the forest environments. Direct natural exposure should not expose participants to natural virtual landscapes (e.g., images or videos of nature projected onto the screen or viewed using virtual reality goggles) but to outdoor environments containing forest elements.
Comparison	Studies including at least one control group
Outcomes	Any outcome related to depression or anxiety
Setting	Studies of environments that primary study authors described as a forest
Study design	Randomized controlled trials (RCTs) or quasi-RCTs

**Table 3 ijerph-18-12685-t003:** Modulator effect analyses for the effects of forest therapy on depression.

Subgroups	Subgroup by Factors	k	Pooled Hedges’s g (95% CI)	Test of Null		Heterogeneity		
				Z-Value	*p*-Value	Q-Value	df (Q)	*p*-Value
Participant	Overall	23	1.133 (−1.490 to −0.776)	−6.220	<0.001	5.793	3	0.122
	Chronic disease	3	1.009 (−1.974 to −0.043)	−2.048	0.041			
	Healthy	10	1.242 (−1.783 to −0.701)	−4.501	<0.001			
	Infants & children & adolescent	3	0.136 (−1.082 to 0.809)	−0.282	0.778			
	Mental disorders	7	1.522 (−2.190 to −0.854)	−4.466	<0.001			
Activity type	Overall	23	1.139 (−1.513 to −0.765)	−5.965	<0.001	0.085	1	0.770
	Day	7	1.221 (−1.888 to −0.554)	−3.587	<0.001			
	Session	16	1.101 (−1.553 to −0.649)	−4.775	<0.001			
Activity content	Overall	22	1.148 (−1.518 to −0.777)	−6.074	<0.001	3.595	1	0.058
	Forest therapy program	19	1.291 (−1.690 to −0.892)	−6.344	<0.001			
	Forest walking	3	0.252 (−1.249 to 0.745)	−0.496	0.620			
Time	Overall	19	1.018 (−1.393 to −0.644)	−5.324	<0.001	10.418	2	0.005
	Less than 60 min	5	0.356 (−1.089 to 0.376)	−0.953	0.340			
	Within 61 to 120 min	9	0.862 (−1.396 to −0.327)	−3.161	0.002			
	More than 121 min	5	2.035 (−2.790 to −1.279)	−5.279	<0.001			
Duration	Overall	23	1.141 (−1.520 to −0.761)	−5.891	<0.001	3.025	2	0.220
	1~3 weeks	10	1.018 (−1.589 to −0.447)	−3.492	<0.001			
	4~7 weeks	6	1.712 (−2.471 to −0.952)	−4.418	<0.001			
	8~12 weeks	7	0.854 (−1.537 to −0.171)	−2.452	0.014			

**Table 4 ijerph-18-12685-t004:** Modulator effect analyses for the effects of forest therapy on anxiety.

Subgroups	Subgroup by Factors	k	Pooled Hedges’s g (95% CI)	Test of Null		Heterogeneity		
				Z-Value	*p*-Value	Q-Value	df (Q)	*p*-Value
Participant	Overall	7	2.184 (−3.249 to −1.119)	−4.019	<0.001	2.644	1	0.104
	Chronic disease	3	3.236 (−4.892 to −1.580)	−3.829	<0.001			
	Healthy	4	1.442 (−2.832 to −0.051)	−2.032	0.042			
Activity type	Overall	9	1.728 (−2.573 to −0.883)	−4.008	<0.001	3.903	1	0.048
	Day	4	2.711 (−4.001 to −1.420)	−4.118	<0.001			
	Session	5	0.990 (−2.108 to 0.129)	−1.734	0.083			
Time	Overall	7	1.899 (−3.049 to −0.749)	−3.236	0.001	1.733	1	0.188
	Less than 60 min	4	1.242 (−2.752 to 0.268)	−1.612	0.107			
	Within 61 to 120 min	3	2.807 (−4.582 to −1.032)	−3.100	0.002			
Duration	Overall	8	1.901 (−2.886 to −0.916)	−3.782	<0.001	2.731	1	0.098
	Within a week	4	2.753 (−4.164 to −1.342)	−3.824	<0.001			
	2~4 weeks	4	1.091 (−2.467 to 0.285)	−1.554	0.120			

## Data Availability

Not applicable.

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
