# Peer review of "Effect of Forest Therapy on Depression and Anxiety: A Systematic Review and Meta-Analysis"

_ijerph, 2021, doi:10.3390/ijerph182312685_

Round 1

Reviewer 1 Report

L37-38: Please recheck these numbers, they contradict the information in the previous sentence.

L100-103: „female colleage students” is mentioned twice in one sentence, please revise.

L104: Should be „negative emotions” I believe.

L105-106: Why „may reveal”? Should be „have revealed”.

L113-114 need linguistic revision.

L118: I recommend removing „and scientifically”.

https://journals.plos.org/plosone/article?id=10.1371/journal.pone.0172200 can be also cited in the Introduction. Was this study considered for meta-analysis, too?

Figure 1 needs linguistic correction in the parts of excluding studies.

L226: It should be „and for almost all the studies”.

L228: I believe it should be „Format”

Table 3: Digits are not needed for a small sample size like the present one.

Figures 2 and 3 have such a low resolution that the text is basically unreadable. Please improve the resolution.

Section 3.6: “moderating variables”, not “modulating variables”.

L309-310: No participants with chronic diseases are mentioned in the Table 4.

L319-331: I would shorten this text as no moderation effect was presented. In general, there is no need to repeat all numbers which are already mentioned in the Table 4. It is sufficient to mention the trends in general.

L349-350: I would be even more daring here: large effect sizes were evident, not just “significant effect”.

L376-377: “severely exhausted middle-aged male patients”?

L396-416: I suggest shortening this part a bit and avoid overspeculation about the types of forest therapy.

L418: It is “Attention restoration therapy”.

L417-422 should be extended given more details on SRT and ART. Relational Restoration Theory and  Collective Restoration Theory should preferably also be discussed.

L505-507: I would say that objective measures are more reliable and should be used in addition to self-reports.

L538-542: More detailed recommendations would be of great use. The current ones are in no way specific.

L564-565: I would change to „has a large effect on alleviating depression and anxiety symptoms”.

Author Response

Dear Editor and reviewer,

We would like to express our sincere gratitude for your kind consideration and comments on our manuscript. According to reviewers’ comments and suggestions, we revised the manuscript as follows: 

(We marked the revision to the reviewer's comment in blue)

  1. We modified the statistical information (line 35-38).
  2. We modified the repeating word “female college student” (line 99-101).
  3. We modified it to the word “negative emotions” (line 102).
  4. We modified it to the sentence “have revealed” (line 108).
  5. We modified the sentence (line 116-119).
  6. We removed it word “scientifically” (line 120-121).
  7. We cited the article you recommended in the introduction (line 104-107). However, because there were no consequences for depression or anxiety in the body of this article, it could not be included in the meta-analysis.
  8. We corrected the language for the contents of Figure 1 (line 211).
  9. We modified it to the sentence “almost all the studies” (line 230-231).
  10. We modified it word “Format” (line 232).
  11. We deleted Table 3 as you suggested that we don’t need digits because the sample size is small.
  12. We changed Figures 2 and 3 to have high resolution.
  13. We modified it to the word “modulating variables” (line 303).
  14. We inserted the “participants with chronic diseases” in Table 4.
  15. We marked only the subgroups that showed significant differences in the modulating effect analysis and revised the numbers not to repeat (line 304-323).
  16. We modified it to the sentence “forest therapy has large effect sizes, not just significant effect evidence on depression and anxiety” (line 343-344).
  17. We modified it to the word “patients with severe exhaustion disorder” (line 369-370).
  18. We reduced the contents of the text to avoid excessive speculation on the type of forest therapy (line 389-394).
  19. We modified it to the word “Attention Restoration Theory” (line 397).
  20. We extended more detail on SRT and ART. we added Relational Restoration Theory and Collective Restoration Theory to the discussion (Line 396-427).
  21. We added it to the sentence “Future studies should require reliable physiological measures in addition to self-reported questionnaires” (line 514-520).
  22. We added that randomization and blinding must be thorough to do RCT studies. (line 564-572).
  23. We modified it to the sentence “forest therapy has a large effect on alleviating depression and anxiety” (line 605).

Once again, thank you very much for the time spent and the interest shown in this work, and the positive evaluations you have given of it.

Reviewer 2 Report

Ref. Review: IJERPH-1441932

Paper Title:  Effect of Forest Therapy on Depression and Anxiety: A Systematic Review and Meta-Analysis

Dear Authors and Editor,

Based on the text exposed in the entitled paper: "Effect of Forest Therapy on Depression and Anxiety: A Systematic Review and Meta-Analysis", I recommend major revisions before acceptance and publishing on Environmental Research and Public Health.

The work presented in this paper aims to summarize the effects of forest therapy on depression and anxiety using data obtained from randomized controlled trials (RCTs) and quasi-experimental studies.

The idea of this work is original and attends an overall view about the forest therapy effect on mental diseases and indicate the advantages of forests in public health. The methodology for this meta-analysis is complex and is a new approach for systematic reviews involving environmental studies.

Please consider the following recommendations for the improvement of this manuscript:

Material and Methods:

Due to a complex methodology exposed in the manuscript, would you please show all data collection, post-treatment, statistical analysis, and major findings process in a flowchart? Such graphical representation would help to understand all selection decision processes, why some statistical analyses were adopted, and what to expect as an outcome.

Results:

Item 3.4, in the sentence: "The overall effect size was -1.133 (95% CI: -1.491 288 to -0.775, p<0.0001), indicating a high effect size".

The authors are probably informing about the effect size regarding absolute values (the magnitude of a real number without regard to its sign).  I agree about the large or "high" magnitude of effect size, but to be precise, inform in the sentence about the absolute value to avoid misinterpretation.  Would you please do the same for the following results regarding effect size?

Table 4, please align the results of "heterogeneity" with the lines indicated as "overall" from each analyzed category.

Conclusion:

Would you please make more comments about the meta-analyses, PRISMA, and bias analyses in multidisciplinary studies that involve environment and public health content? Does the adopted approach work well, etc.?

Author Response

Dear Editor and reviewer,

We would like to express our sincere gratitude for your kind consideration and comments on our manuscript. According to reviewers’ comments and suggestions, we revised the manuscript as follows:

(We marked the revision to the reviewer's comment in green)

  1. We added a flowchart of the systematic review and meta-analysis process (Figure 1; line 132).
  2. We revised the effect size of the results to an absolute value (line 312-362).
  3. We aligned the results of “heterogeneity” with the lines indicated as “overall” (Table 3; line 325).
  4. We added more comments about the meta-analyses, PRISMA, and bias analyses in multidisciplinary studies that involve the environment and public health content (line 564-572, 590-601).

Once again, thank you very much for the time spent and the interest shown in this work, and the positive evaluations you have given of it.

Reviewer 3 Report

Thank you for giving me the opportunity to review your manuscript. I think topic of your study is so interesting and  forest  therapy will be helpful for improving mental health problem. You attempted to confirm the effect of forest therapy by systematically examining and conducting meta-analysis the effects of forest therapy on depression and anxiety over the past 30 years. You did a hard work. However, I consider it is difficult to publish this SR due to several issues. My comments are as belows:

#1.  The major issue of this review is for conducting meta-analysis. The charateristics in your included studies are too diverse regarding to the participants, interventions, and comparision. Although you conduct subgroup analysis, your meta-analysis do not provide the true intervention effect because these clinical heterogeneity is too large and obvious. And meta-analysis of diverse participants do not provide useful evidence in clinlical setting. Furthermore you included not only RCT but also quasi-RCT. This lead obvious methodological heterogeneity. 

#2. Therefore, I think the rationale for conducting your review is insuffcient because there are recent SRs.

#3. You did not state whether they published a protocol for this review and the protocol registration number was missing. If the authors did not register their protocol, the authors should state why they did not register their protocol in a registry such as PROSPERO. It is important to register their protocol before a systematic review has been done, since ‘Without review protocols, how can we be assured that decisions made during the research process aren’t arbitrary, or that the decision to include/exclude studies/data in a review aren’t made in light of knowledge about individual study findings? (http://www.prisma-statement.org/Protocols/WhyProtocols)’. And you also should present PRISMA checklist as an Appendix. 

#4. You should conduct more comprehensive searching. You did not search using Chinese and Japanese database. 

#5. The manuscipt is not readable. The introduction and discussion section has too long parapraph. Many sentence has no relevant reference.

Author Response

Dear Editor and reviewer,

We would like to express our sincere gratitude for your kind consideration and comments on our manuscript. According to reviewers’ comments and suggestions, we revised the manuscript as follows:

(We marked the revision to the reviewer's comment in red)

  1. As the reviewer said, we are aware of the high heterogeneity of effect sizes. Therefore, the effect size model was designated as a random effect, and we addressed these issues in the limitations. In addition, we included NRCT studies and RCT studies in meta-analysis, and this is still a lack of RCT research in forest therapy research. The previous meta-analysis related to prior forests and health effects included RCT and NRCT, such as crossover design and quasi-experimental designs [Kotera, Richardson, Sheffield, 2020; Rosa, Larson, Collado, Profice, 2021; Stier-Jarmer, Theroner, Kirschneck et al., 2021].

Still, we included only RCTs and quasi-experimental designs to reduce the heterogeneity of research methodology as much as possible.

We added these problems with heterogeneity to the limitation (line 503-509).

  1. We did not preregister the study in PROSPERO. We only recognized that pre-registration was not mandatory and proceeded with literature extraction. However, at the same time, the authors recognize the importance of registering systematic reviews in PROSPERO and, from now on, it will be an essential step for us each time we initiate a review.

We specified in the discussion the failure to register in advance for this review and the need to register protocols in future studies (line 547-552).

Also, we presented the PRISMA checklist in the appendix (line 629).

  1. We could not conduct literature searches in Japanese or Chinese databases, conducting much research on forest therapy. Thus, we may have missed a systematic review. However, since we used four major international and two major Korean databases, we identified many for this systematic review and meta-analysis.

We added these problems in the discussion (line 534-538).

  1. We think the introduction and discussion are prolonged because we have addressed why forest therapy effectively improves depression and anxiety.

We added references related to many sentences to this review.

Finally, we hope that you will rethink your decision, especially considering the positive evaluations received by other reviewers and that, after the corrections made (both by your advice and that of the other reviewers) the quality of the manuscript has improved significantly.

Once again, thank you very much for the time spent and the interest shown in this work, and the positive evaluations you have given of it.

Reviewer 4 Report

The manuscript entitled "Effect of Forest Therapy on Depression and Anxiety: A Systematic Review and Meta-Analysis" aimed to summarize the effects of forest therapy on depression and anxiety using data obtained from randomized controlled trials and quasi-experimental studies.

The review of the scientific literature is done very well: the research is scrupulous, the explanation is precise and the results interesting. Here are some doubts, certainly minor, that I would like to point out in order to improve the paper.

Materials and Methods
Pag. 3, line 123: It is reported that the Preferred Reporting Items for Systematic reviews and Meta-Analyzes (PRISMA) guidelines refer to bibliographic entry number 20, while the correct one is entry number 30 (Moher, Librati, Tetzlaff & Altman, 2009). Definitely a typo, to correct.
Pag. 3, line 126: Among the inclusion criteria of the papers is indicated the inclusion of studies published from January 1980 to December 2020. Specify why this time range was chosen.

Results
Pag. 5, line 194: Both in figure 1 and in the text I read that the papers initially identifying was 163,754, with the relative specification of the various databases. It is indicated that: Scopus, n = 20.711; PubMed, n = 5,777; MEDLINE, n = 352; 194 Web of science, n = 14,138; RISS, n = 5,277; DBpia, n = 117,479. The sum of these papers, however, is not 163,754, but 163,734. Perhaps there is a mistake or confusion as to why those 20 missing papers were not included. Review this paragraph.

Discussion
Pag. 12, line 388: In the manuscript it is written that the results of review were different from those of previous studies. It is better to specify which studies are being referred to.
Pag. 13, line 468: Although the discussion is well written, I believe it can be improved in some points, integrating aspects from different cultural realities, to support its results. Eg. Gori, Giannini, Topino et al., 2021; Burgassi, Giannini, Giovannelli, et al., 2017; Mishra, Srivastava, Tiwary et al., 2018. I believe that a more complete scientific framework can offer more insights on an international level.

Author Response

Dear Editor and reviewer,

We would like to express our sincere gratitude for your kind consideration and comments on our manuscript. According to reviewers’ comments and suggestions, we revised the manuscript as follows:

(We marked the revision to the reviewer's comment in purple)

  1. We modified it to entry number 31 (line 125).
  2. We made a mistake in the search period (from January 1980 to December 2020). So, we revised it to “from January 1990 to December 2020”. And we added the reason for selecting the search period from 1990 to 2020. (line 128-130).
  3. We made a mistake in the sum calculation of the paper initially identifying the sum. So, we revised it to 163,734 records. (line 199).
  4. We added references from the previous study. (line 381-388).
  5. We added that we should conduct research considering other aspects of cultural reality in the discussion (line 473-479).

Once again, thank you very much for the time spent and the interest shown in this work, and the positive evaluations you have given of it.

Round 2

Reviewer 2 Report

The manuscript can be accepted for publication in the International Journal of Environmental Research and Public Health. 

Author Response

We would like to express our sincere gratitude for your kind consideration and acceptance of our manuscript.

Reviewer 3 Report

Although authors did make a effort, I think some issues should be addressed. 

Major issues 
- 'Embase' together with previous database should be searched for comprehensive reviews based on general recommendation and previous relevant review. 
(https://training.cochrane.org/handbook/current/chapter-04#section-4-4
https://www.bmj.com/content/372/bmj.n71)

- Conduct to subgroup anaylsis according to the duration and control intervention, additionaly. And you should anaylsis each depression and anxiety in the subgroup anaylsis.  Depression and anxiety are diffrent each other.

- Chcek ROB.  You judged that RCTs were rated as low ROB for Deviation from intended interventions. However. it is impossible to conduct blinding of participants and personnel in your 5 RCTs.

Minor issues
-introduction: This  sentence in below is wrong. Correct this and provide appropriate citation. and "both antidepressants and anti-anxiety drugs and psychotherapy are ineffective in the short term."

-introduction: Antidepressants and anti-anxiety drugs has diffrent mechasnism and properties. Provide each information of adverse event, dependency and misuse and so on regading each drugs. And add the rationale of non-pharmacological treatment. 

-introduction:  Add the limitation of psychotherapy as non-pharmacological treatment for depression and anxiety to stress the rationale of your study.

- The authors cited the PRISMA 2009 statement, but given appendix, it appears they actually used the 2020 statement.

- Present the full search strategies for all databases. Provide specific search terms used in each database to help other researcher to conduct same searching, not just at least one database.  

- Table A1. Search terms used for literature searches. 
Check and correct thess errors. 
26. 2 OR 3 OR 4 OR 5 OR 6 OR 7 OR 8 OR 9 OR 10 OR 11 OR 12 OR 13 OR 14 OR 15 OR 16 OR 17 OR 18 OR 19
39. 21 OR 22 OR 23 OR 24 OR 25
40. 26 AND 38
38. Mental health

Author Response

Dear Editor and reviewers,

We would like to express our sincere gratitude for your kind consideration and comments on our manuscript. According to reviewers’ comments and suggestions, we revised the manuscript as follows:

(We marked the revision to the reviewer's comment in red)

  1. We searched ‘Embase’ together with the previous database. However, there was no change in the final studies for this review (line 203-218).
  2. We modified subgroup analysis (line 313-352).
  3. We confirmed the ROB 2.0. As in your comment, it is impossible to blind participants and personnel in five RCTs, but we judged as low ROB for Deviation from intended intervention because there was no deviation from the intended intervention due to the clinical trial context and appropriate analysis was used to estimate the effect of the assignment.
  4. We modified and added the side effects of anti-anxiety drugs and antidepressants, and evidence of non-pharmacological treatment in the introduction (line 57-69).
  5. We modified Reference to cite the PRISMA 2020 (line 129, Reference [45]).
  6. We added full search strategies and specific search terms used in each database in Appendix A. (Table A1) (line 657-658).
